# TriORM: Workload-Aware Neural–Symbolic Multi-Objective Optimization for ORM Mapping Design

### Sasan Azizian
College of Science and Technology, Bellevue University
Bellevue, Nebraska, USA
sazizian@bellevue.edu

### Artin Azizian
School of Computer Science, McGill University
Montreal, Quebec, Canada
artin.azizian@mail.mcgill.ca

### Ayoub Hazrati
The Vanguard Group
Valley Forge, Pennsylvania, USA
hazrati.ayoub@gmail.com

### Elham Rastegari
Business Intelligence & Analytics, Creighton University
Omaha, Nebraska, USA
elhamrastegari@creighton.edu

## ABSTRACT

Object-relational mapping (ORM) frameworks simplify persistence, yet routine mapping choices—including inheritance strategy, association encoding, and denormalization—can substantially alter the generated SQL and the trade-offs among query latency, insert/update cost, and storage footprint. Existing optimization approaches either guarantee semantic validity but incur expensive per-candidate deployment and benchmarking, or learn from schema structure alone and therefore miss the fact that workload behavior is mapping-dependent. We present TriORM, a workload-aware neural–symbolic framework for multi-objective ORM mapping design that removes per-candidate workload execution from the *online* recommendation loop while preserving validity by construction. TriORM (i) enumerates admissible mappings via bounded relational synthesis in Alloy, (ii) concretizes an abstract workload into schema-specific SQL templates for each candidate, and (iii) predicts continuous objectives using a tri-input model that fuses a typed schema-graph encoder, a concretized-workload encoder, and interpretable static cost proxies. The resulting predictions enable Pareto filtering and user-weighted selection without deploying or executing each candidate at inference time; profiling is performed only *offline* to obtain supervision. On nine TradeMaker/Leant benchmark models, TriORM improves mean Pareto-front approximation over Leant (GD/HV 0.03/0.78 vs. 0.07/0.61) and reduces end-to-end recommendation time ($3.6{\times}10^3$ s vs. $2.6{\times}10^4$ s), while preserving semantic correctness within the chosen synthesis bounds.

## CCS CONCEPTS

• **Software and its engineering → Formal software verification**.

## KEYWORDS

Object–Relational Mapping, Performance Prediction, Neural–Symbolic Systems, Multi-objective Optimization, Schema Synthesis, Regression

**ACM Reference Format:**
Sasan Azizian, Ayoub Hazrati, Artin Azizian, and Elham Rastegari. 2026. TriORM: Workload-Aware Neural–Symbolic Multi-Objective Optimization for ORM Mapping Design. In *Proceedings of 3rd ACM International Conference on AI-powered Software (AIware '26)*. ACM, New York, NY, USA, 9 pages. https://doi.org/XXXXXXX.XXXXXXX

## 1 INTRODUCTION

Relational database management systems (RDBMSs) remain the default substrate for persistent data, while application logic is commonly expressed in object-oriented abstractions. Object–relational mapping (ORM) frameworks (e.g., Hibernate, Django ORM, SQLAlchemy, Entity Framework) mediate this impedance mismatch by mapping an object model to a relational schema and compiling application-level operations into SQL [1, 2, 4, 13, 16, 24]. In modern systems, however, ORM is not merely a convenience layer: *mapping design*—inheritance encoding (single-table, joined-subclass, table-per-class), association representation (foreign key vs. join table), denormalization, and indexing conventions—determines the physical layout and the concrete SQL/transaction patterns executed by the DBMS. These choices reshape join topology, redundancy, constraint and index maintenance, and update behavior, and can produce order-of-magnitude differences in query latency, write cost, and storage footprint even when application semantics are preserved [4, 12, 16, 21].

A central challenge is that ORM design is both *combinatorial* and *workload-induced*. Even moderate object models admit many correctness-preserving realizations: each inheritance edge and association can be encoded in multiple ways while satisfying integrity and multiplicity constraints [12, 16, 21]. More subtly, the effective database workload depends on the chosen mapping. The same application operation (e.g., "load a customer with orders" or "create an order") can compile to materially different SQL across candidates: joined-subclass introduces additional inheritance joins; join-table associations add extra hops and referential checks; single-table removes joins but reads wider tuples and enforces predicates via discriminators; denormalization can reduce join depth but increases write sets and index/constraint maintenance. As a result, cost is not

a function of schema structure alone. Accurate ranking requires reasoning about the physical schema *and* the SQL workload it induces.

Prior work captures important parts of this space, but leaves a practical gap for fast, workload-specific recommendations under semantic constraints. Symbolic synthesis can guarantee validity by construction by enumerating only constraint-satisfying schemas (e.g., via Alloy), yet the candidate space grows combinatorially with model size and interacting mapping choices [11, 20]. Benchmarking-driven optimizers can measure accurate costs, but require per-candidate deployment, data loading, and workload execution—a dominant bottleneck when hundreds to thousands of mappings must be explored [9]. Learning-based approaches improve scalability, but schema-only representations can misrank candidates precisely because they ignore mapping-induced workload realization. What is missing is a design-time optimizer that simultaneously (i) preserves semantic validity, (ii) makes workload dependence explicit and comparable across mappings, and (iii) removes per-candidate execution from the recommendation loop.

We present TriORM, a neural–symbolic framework for workload-aware, multi-objective ORM mapping design. TriORM separates *validity* from *performance reasoning*. First, it compiles the object model and admissibility constraints into a bounded relational specification and enumerates only valid candidate schemas using Alloy-based synthesis [11, 20]. Second, it *concretizes* an abstract workload (operation templates with frequencies) into schema-specific SQL/transaction templates for each candidate, capturing mapping-induced variability while preserving semantic intent across candidates. Third, it predicts continuous objective values without per-candidate benchmarking at inference time using a tri-input predictor that fuses: (i) a typed schema-graph encoder capturing relational structure and physical design elements; (ii) a concretized-workload encoder capturing join and write patterns of the realized SQL; and (iii) compact, interpretable cost proxies capturing join pressure and write amplification. The resulting predictions support Pareto filtering or user-weighted selection over $(T_{qry}, T_{ins}, M)$ under standard multi-objective criteria [19, 31]. Expensive measurements are performed only *offline* to obtain supervision and evaluation ground truth, not as part of the inference-time recommendation loop. **Practical use.** TriORM is intended for design-time guidance. Given an object model and admissibility constraints, developers provide a lightweight workload sketch (operation templates and approximate frequencies). TriORM returns a small predicted Pareto set of *valid* mappings (or a single mapping under user weights), enabling early exploration of read/write/footprint tradeoffs before costly end-to-end deployment and profiling.

**Contributions. (1) Neural–symbolic pipeline for workload--aware ORM optimization:** a two-stage design that (A) enumerates only admissible mappings via bounded relational synthesis and (B) ranks them using workload-conditioned prediction without per-candidate execution in the recommendation loop. **(2) Schema–conditioned workload concretization:** a deterministic procedure that translates abstract operation templates into schema-specific SQL/transaction templates per candidate, making workload dependence explicit and comparable across mappings. **(3) Tri-input continuous predictor for multi-objective tradeoffs:** a fused representation of typed schema graphs, concretized SQL workloads, and interpretable cost proxies to predict $(T_{qry}, T_{ins}, M)$ for Pareto-aware recommendation. **(4) Empirical evaluation on standard benchmarks:** results on the nine-model TradeMaker/Leant suite showing improved Pareto quality and substantially lower end-to-end recommendation cost relative to representative baselines.

## 2 PROBLEM FORMULATION AND OVERVIEW

ORM mapping design is a *validity-constrained, workload-conditioned* multi-objective optimization problem. A mapping must preserve object-model semantics and integrity constraints, yet the best valid mapping depends on the expected workload and the trade-off among query latency, write cost, and storage footprint. TriORM addresses this setting by (i) enumerating only valid candidates and (ii) ranking them with workload-aware performance prediction.

*Object model and constraints.* We model the application domain as a typed object model $O = (C, \mathcal{A}, \mathcal{R}, \mathcal{H})$, where $C$ is the set of classes, $\mathcal{A} : C \to 2^{\text{Attr}}$ assigns typed attributes, $\mathcal{R}$ is the set of associations with multiplicities, and $\mathcal{H} \subseteq C \times C$ is an inheritance relation [4, 16, 21]. Let $\Phi$ denote admissibility constraints over ORM realizations, including allowed inheritance strategies (single-table, joined-subclass, table-per-class), allowed association encodings (FK vs. join table), and relational integrity requirements (key uniqueness, referential integrity, multiplicity preservation) [12, 16, 21].

*Valid schemas and design space.* A candidate relational schema is $S = (\mathcal{T}, \mathcal{K}, \mathcal{F})$, where $\mathcal{T}$ denotes tables/columns, $\mathcal{K}$ primary/unique keys, and $\mathcal{F}$ foreign keys. We call $S$ *valid* iff

$$S \models (O, \Phi).$$

The feasible design space is

$$\mathcal{S}(O, \Phi) = \{ S \mid S \models (O, \Phi) \},$$

which can be large because each inheritance edge and association admits multiple correctness-preserving encodings, and their decisions interact [11, 20].

*Workload dependence and objectives.* Let $W_{\text{abs}} = \{(op_j, f_j)\}_{j=1}^{n}$ be an abstract workload profile, where $op_j$ is an application-level operation template and $f_j$ is its frequency; in our experiments, operation templates and approximate frequencies are taken from the TradeMaker/Leant benchmark workloads and normalized such that $\sum_j f_j = 1$. Crucially, the *realized* SQL workload depends on the mapping: the same $op_j$ can compile into different SQL shapes under different inheritance/association encodings [12, 16, 21]. We therefore treat performance as *schema-conditioned*: each schema $S$ induces a realized workload $W_S$. For each valid schema $S$, we define the objective vector

$$Y(S) = \big(T_{\text{ins}}(S), \ T_{\text{qry}}(S), \ M(S)\big),$$

where $T_{\text{ins}}$ is insert/update latency, $T_{\text{qry}}$ is query latency, and $M$ is storage footprint.

*Pareto-optimality and challenges.* We seek Pareto-efficient schemas

$$\mathcal{P}^{\star} = \{ S \in \mathcal{S}(O, \Phi) \mid \nexists S' \in \mathcal{S}(O, \Phi) : Y(S') \prec Y(S) \},$$

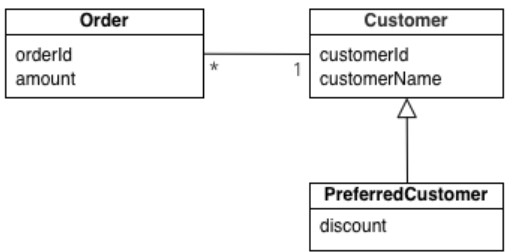

Figure 1: Object model (`Customer–Order`) with inheritance and a 1:*N* association.

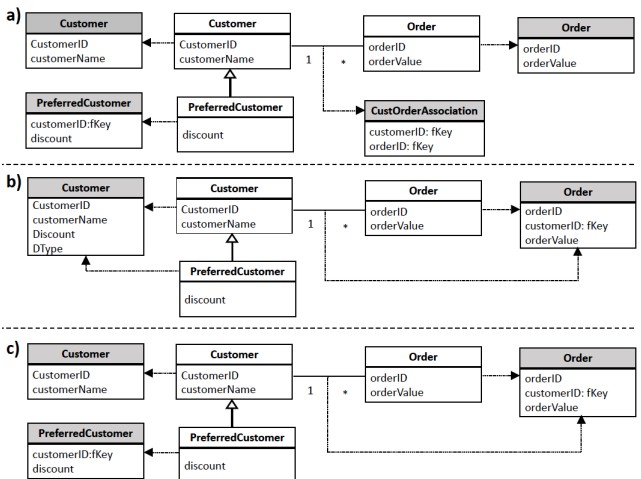

Figure 2: Representative valid mappings for Fig. 1. Mapping choices change join topology, redundancy, and constraint/index structure.

where $\prec$ denotes Pareto dominance; we evaluate tradeoff quality using standard metrics such as GD and HV [19, 31]. The problem is challenging because (i) $\mathcal{S}(O, \Phi)$ grows combinatorially with interacting mapping choices, and (ii) obtaining $Y(S)$ via schema deployment, data loading, and executing $W_S$ is expensive at scale [9]. These constraints motivate ranking valid candidates via workload-aware prediction rather than per-candidate benchmarking in the inference-time loop.

## 3 ILLUSTRATIVE EXAMPLE

We use a minimal `Customer–Order` domain to motivate two design requirements for TriORM: (i) even small object models admit many *semantics-preserving* ORM mappings with materially different physical layouts, and (ii) performance must be modeled as *schema-conditioned* because the realized SQL workload depends on the mapping. Figure 1 shows an object model with an inheritance edge `PreferredCustomer → Customer` and a 1:*N* association `Customer → Order`. Under standard mapping rules, two choices largely determine the relational realization (Fig. 2): *inheritance encoding* (single-table, joined-subclass, table-per-class) and *association encoding* (foreign key vs. join table) [4, 12, 16, 21]. All mappings in Fig. 2 are *valid*—they preserve object-model semantics and integrity constraints—yet they induce different join connectivity, redundancy/nullability patterns, and constraint/index topology.

**Why workload realization matters.** The same application-level operation can compile to different SQL depending on the mapping. For instance, getCustomerWithRecentOrders(id) typically requires an inheritance join under joined-subclass but not under single-table; likewise, representing the 1:*N* association via a join table introduces an extra hop (and corresponding checks) compared to an FK encoding [12, 16, 21]. Therefore, the realized SQL template $q^{(S)}$ is a function of the chosen schema $S$, and predictors that ignore this dependence (e.g., schema-only models) can systematically misrank semantically valid mappings. **Mapping-induced multi-objective tradeoffs.** These structural differences directly shift the tradeoff among query latency, write cost, and footprint ($T_{qry}, T_{ins}, M$). More normalized layouts often increase join depth (penalizing reads), whereas denormalized or single-table designs can reduce joins but increase write amplification and footprint due to redundancy and index/constraint maintenance [4, 12, 16]. This example motivates TriORM's approach: enumerate only valid mappings, *realize* the workload per mapping, and rank candidates using workload-conditioned prediction.

## 4 APPROACH

TriORM treats ORM mapping design as a *validity-constrained, workload-conditioned* multi-objective optimization problem. Given an object model $O$, admissibility constraints $\Phi$, and an abstract workload profile $W_{abs} = \{(op_j, f_j)\}_{j=1}^{n}$ (operation templates with frequencies, $\sum_j f_j=1$), the goal is to recommend *valid* schemas that expose favorable tradeoffs among query latency $T_{qry}$, insert/update latency $T_{ins}$, and footprint $M$. The core technical challenge is that the *realized* SQL workload is *mapping-dependent*: the same application operation can compile into different SQL under different inheritance/association encodings, changing join topology, access paths, and write amplification [4, 12, 16, 21].

*Design principle: separate validity from ranking.* TriORM separates: **(i) semantic validity**—generate only schemas that preserve object-model semantics and satisfy mapping/integrity constraints—from **(ii) performance reasoning**—rank *already-valid* schemas under a workload-specific tradeoff. This avoids the failure mode of proposing high-scoring but invalid mappings: learning is used only after symbolic validity is guaranteed [11, 20].

*Two-stage pipeline (overview).* Figures 3a–3b summarize TriORM's two-stage design. **Stage A (symbolic)** enumerates a finite set of valid schemas and deterministically *realizes* the abstract workload under each schema. **Stage B (learning)** predicts continuous objective values for each candidate from three aligned inputs: **I1** schema structure, **I2** realized workload, and **I3** static cost proxies. The output is either (i) a predicted Pareto set of valid schemas or (ii) a single schema selected via user weights over objectives.

### 4.1 Stage A: Validity-Preserving Synthesis and Workload Realization

*A.1 Bounded synthesis of valid schemas.* We compile $(O, \Phi)$ into a bounded relational specification $A \leftarrow \text{Trans}(O, \Phi)$ and enumerate candidates $\mathcal{S} = \text{Synth}(A) = \{S_1, \ldots, S_K\}$ such that every $S_i$ is valid by construction: $S_i \models (O, \Phi)$. Alloy encodes mapping and integrity constraints in relational logic and enumerates satisfying instances via SAT-based bounded solving [20]. This paradigm has been used

effectively for correctness-preserving ORM enumeration in prior work [11].

*A.2 Alloy intuition (simplified).* Listing 1 illustrates the style of constraints we encode (e.g., key uniqueness and multiplicities). The full specification additionally captures inheritance strategies, association encodings (including join tables), column placement/nullability, and schema well-formedness; the key invariant is that synthesis returns only constraint-satisfying schemas.

**Listing 1: Alloy intuition: enforcing key uniqueness and a $1{:}N$ association as a typed reference (simplified).**

```
// Each relation is modeled as a set of rows (tuples).
sig CustomerRow { id: one Int } // primary key column
sig OrderRow { customer: one CustomerRow } // FK column as typed reference

// Primary-key uniqueness (simplified).
fact CustomerPKUnique {
  all disj c1, c2: CustomerRow | c1.id != c2.id
}

// $1{:}N$ multiplicity: each OrderRow references exactly one CustomerRow.
// (Multiple orders may reference the same customer.)
fact OneToManyCustomerOrder {
  all o: OrderRow | one o.customer
}
```

*A.3 Schema-conditioned workload realization (concretization).* The abstract workload $W_{abs}$ is schema-agnostic: it specifies operations in application terms (e.g., getCustomerWithRecentOrders(id)), not SQL. For each candidate $S_i$, TRIORM deterministically instantiates $W_{abs}$ into a schema-specific realized workload:

$$W_i = \text{Concretize}(W_{abs}, S_i) = \{(q_j^{(S_i)}, f_j)\}_{j=1}^n,$$

where each $q_j^{(S_i)}$ is a canonical SQL (or transaction) template implementing $op_j$ under schema $S_i$. Concretization resolves inheritance encoding, association representation, and denormalized attribute placement, producing templates that are *semantics-equivalent across candidates* yet structurally different (tables touched, join depth, predicate placement, write sets) [12, 16, 21]. These mapping-induced differences are precisely what must be modeled for workload-aware ranking.

*Stage-A artifacts and provenance.* Stage A yields two artifacts: **O1**: valid schemas $\{S_i\}_{i=1}^K$; and **O2**: realized workloads $\{W_i\}_{i=1}^K$ with $W_i = \text{Concretize}(W_{abs}, S_i)$. Stage B consumes them as: **I1** schema graph derived from **O1**, **I2** realized workload from **O2**, and **I3** cost proxies computed from **I1**+**I2** via $\phi(G_{S_i}, W_i)$.

## 4.2 Stage B: Tri-Input Encoding and Continuous Multi-Objective Prediction

For each candidate schema $S_i$, TRIORM predicts a continuous objective vector

$$\widehat{Y}(S_i) = (\widehat{T}_{qry}(S_i), \widehat{T}_{ins}(S_i), \widehat{M}(S_i)).$$

*Model configuration.* Unless otherwise stated, we use an $L{=}2$ layer R-GCN for $G_{S_i}$ with hidden size $d{=}128$ and mean+max pooling for readout. For SQL templates, we use a lightweight Transformer encoder with 2 layers, 4 attention heads, and hidden size $d{=}128$. Static proxies are encoded by a 2-layer MLP (hidden size 128). The fused representation is passed through a shared 2-layer trunk (hidden size 256) with three regression heads. We regress latencies and footprint in log-space and train with Huber loss.

*Input I1 (from O1): typed schema graph $G_{S_i}$.* We represent each synthesized schema as a typed directed graph $G_{S_i} = (V, E, \tau_V, \tau_E, X)$. Nodes encode physical entities (e.g., TABLE, COLUMN, INDEX); edges encode typed relations (e.g., table–column containment, PK/FK connectivity, index coverage, and mapping-induced inheritance/association links). Node features capture lightweight properties (datatype class, nullability, key flags, index arity/uniqueness, and table-width proxies). We encode $G_{S_i}$ using an R-GCN [27]:

$$h_v^{(\ell+1)} = \sigma\Big(W_0^{(\ell)} h_v^{(\ell)} + \sum_{r \in \mathcal{R}} \sum_{u \in \mathcal{N}_r(v)} \frac{1}{c_{v,r}} W_r^{(\ell)} h_u^{(\ell)}\Big),$$

where $\mathcal{N}_r(v)$ are $r$-typed neighbors and $c_{v,r}$ normalizes fan-in. After $L$ layers, we obtain a schema embedding by pooling node states:

$$z_S(S_i) = \text{Readout}\Big(\{h_v^{(L)} \mid v \in V\}\Big).$$

*Input I2 (from O2): realized workload $W_i$ and workload embedding $z_W(S_i)$.* Stage A provides a realized workload $W_i = \{(q_j^{(S_i)}, f_j)\}_{j=1}^n$ per schema. Before encoding, we canonicalize each template to remove superficial variance (formatting, identifier names, literals) while preserving cost-relevant structure (join graph, predicate shape, grouping/sorting operators, and write sets). We encode each canonical template with a Transformer $E_Q$ [30] and aggregate by workload frequency:

$$e_j^{(S_i)} = E_Q(q_j^{(S_i)}), \qquad z_W(S_i) = \sum_{j=1}^n f_j \cdot e_j^{(S_i)}.$$

*SQL canonicalization.* We canonicalize templates by (i) normalizing whitespace and keyword casing; (ii) renaming schema-specific identifiers to deterministic placeholders (e.g., T1, C3); (iii) replacing literals with typed parameters; (iv) deterministically ordering commutative predicate conjuncts; and (v) using a fixed join serialization. This preserves structural cost signals while reducing spurious lexical variance.

*Input I3 (from I1+I2): static cost proxies $x_F(S_i)$.* We compute lightweight, interpretable proxies $x_F(S_i) = \phi(G_{S_i}, W_i)$ that approximate dominant cost drivers without executing the workload, including schema counts (#tables, #FKs, #indexes), join-depth proxies, predicate-to-index coverage indicators, and write-amplification proxies (e.g., index updates per insert). An MLP maps $x_F(S_i)$ to an embedding $z_F(S_i)$.

*Fusion and regression.* We fuse modality embeddings [5, 6] via concatenation and simple interactions:

$$z_i = \text{Fuse}(z_S, z_W, z_F) = [z_S;\ z_W;\ z_F;\ z_S \odot z_W;\ |z_S - z_W|],$$

and predict objectives with a shared trunk and objective-specific heads: $\widehat{Y}(S_i) = g_\theta(z_i)$. We train a multi-task objective in log-space:

$$\mathcal{L} = \lambda_q\, \ell(\widehat{\log T}_{qry}, \log T_{qry}) + \lambda_i\, \ell(\widehat{\log T}_{ins}, \log T_{ins}) + \lambda_m\, \ell(\widehat{\log M}, \log M),$$

where $\ell$ is Huber loss unless noted otherwise.

*Inference-time recommendation and selection.* At recommendation time, TRIORM performs *no per-candidate schema deployment or workload execution*. Given $(O, \Phi, W_{abs})$, it runs bounded synthesis, concretization, feature extraction, and prediction, then returns either: (i) the predicted Pareto set $\text{Pareto}(\{\widehat{Y}(S_i)\})$, or (ii) a weighted selection:

$$S^\star = \arg\min_{S_i \in \mathcal{S}} \alpha\, \widehat{T}_{ins}(S_i) + \beta\, \widehat{T}_{qry}(S_i) + \gamma\, \widehat{M}(S_i), \qquad \alpha, \beta, \gamma \geq 0.$$

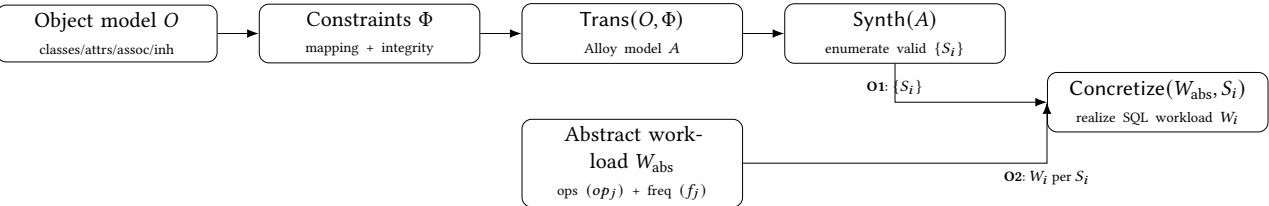

(a) Stage A (symbolic). Outputs: O1 valid schemas $\{S_i\}$ and O2 realized workloads $W_i$.

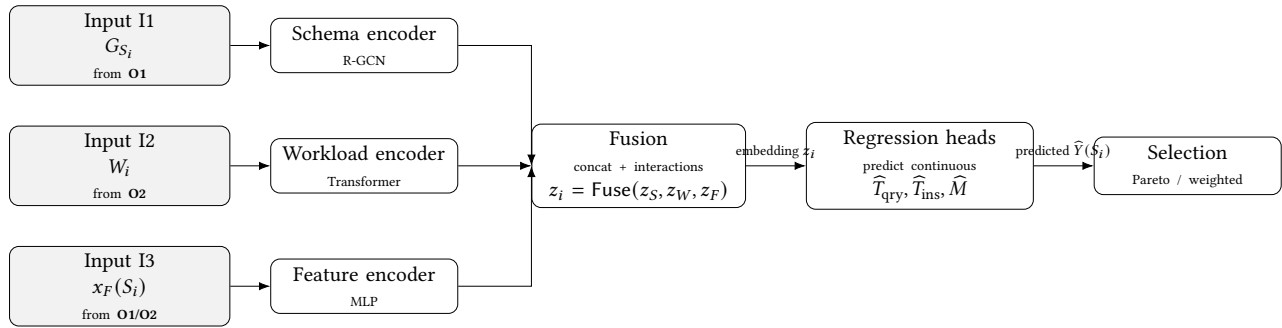

(b) Stage B (learning). Inputs explicitly originate from Stage A outputs (O1, O2).

**Figure 3: TriORM pipeline with explicit dataflow. Stage A synthesizes valid schemas and realizes a schema-specific SQL workload (O1, O2). Stage B consumes these outputs as inputs (I1/I2/I3) to predict continuous objectives for Pareto or weighted selection.**

---

**Algorithm 1** TriORM inference-time recommendation

**Require:** Object model $O$, constraints $\Phi$, abstract workload $W_{abs} = \{(op_j, f_j)\}_{j=1}^n$,
 trained predictor Pred, optional weights $(\alpha, \beta, \gamma)$
 $A \leftarrow \text{Trans}(O, \Phi)$
 $\mathcal{S} \leftarrow \text{Synth}(A)$          ▷ **O1:** valid schemas $\{S_i\}$
 **for all** $S_i \in \mathcal{S}$ **do**
  $W_i \leftarrow \text{Concretize}(W_{abs}, S_i)$     ▷ **O2:** realized workload for $S_i$
  $G_{S_i} \leftarrow \text{Graphify}(S_i)$          ▷ **I1** from O1
  $x_F(S_i) \leftarrow \phi(G_{S_i}, W_i)$        ▷ **I3** from I1+I2
  $\widehat{Y}(S_i) \leftarrow \text{Pred}(G_{S_i}, W_i, x_F(S_i))$
 **end for**
 **if** $(\alpha, \beta, \gamma)$ is provided **then**
  **return** $\arg\min_{S_i \in \mathcal{S}} \alpha \widehat{T}_{ins}(S_i) + \beta \widehat{T}_{qry}(S_i) + \gamma \widehat{M}(S_i)$
 **else**
  **return** $\text{Pareto}(\{\widehat{Y}(S_i)\})$
 **end if**

---

Algorithm 1 summarizes the inference-time procedure.

## 5 EVALUATION

We evaluate TriORM on the standard ORM-optimization benchmark suite to answer two questions: (1) can TriORM recover high-quality *measured* tradeoffs among $(T_{qry}, T_{ins}, M)$ while recommending only *valid* mappings, and (2) can it reduce the inference-time cost of optimization by removing per-candidate workload execution from the recommendation loop? We organize the evaluation around four research questions (RQs) and report both recommendation quality and end-to-end efficiency.

### 5.1 Research Questions

**RQ1 (Pareto quality).** How closely does TriORM's predicted non-dominated set approximate the *measured* Pareto frontier over $(T_{qry}, T_{ins}, M)$?

**Table 1: Benchmark subjects (TradeMaker/LEANT). Model sizes.**

| Model | #Cls | #Asc | #Inh |
|---|---|---|---|
| Customer–Order [11] | 4 | 2 | 1 |
| Decider [9] | 10 | 11 | 5 |
| E-commerce [22] | 15 | 9 | 7 |
| CSOS [11] | 14 | 4 | 6 |
| Bank Mgmt. [26] | 9 | 7 | 3 |
| Camping [28] | 8 | 6 | 2 |
| Flagship [9] | 13 | 10 | 8 |
| Online Store [22] | 12 | 8 | 4 |
| Library Mgmt. [14] | 11 | 7 | 4 |

**RQ2 (Pareto-set identification).** How accurately does TriORM identify Pareto-optimal candidates (Precision/Recall/F1) with respect to the measured frontier?

**RQ3 (End-to-end efficiency).** What are the end-to-end runtime and peak memory footprint of TriORM compared to benchmark-driven baselines?

**RQ4 (Input contribution).** Which inputs contribute most to accurate prediction and Pareto recommendation: schema graph (I1), concretized workload (I2), or static cost proxies (I3)?

### 5.2 Experimental Setup

*Subjects.* We use the nine object models from the TradeMaker/LEANT benchmark suite [9, 11]. We follow the same model definitions and workload templates as prior work for comparability. Table 1 reports model statistics that correlate with synthesis-space size and the difficulty of workload-conditioned ranking.

*Candidate generation and validity.* For each subject, we compile $(O, \Phi)$ into an Alloy specification and enumerate candidate schemas via bounded synthesis. By construction, every candidate satisfies $S \models (O, \Phi)$ [11, 20]. Across the nine subjects, synthesis yields 100–2700 valid schemas per model under our scopes, and we obtain measurements for ~9,500 schemas in total.

*Offline profiling and ground truth.* We obtain ground-truth labels *offline* by materializing each schema in PostgreSQL and executing a controlled profiling protocol (schema creation, data loading, warm-up, and repeated runs of benchmark workload templates). We record query latency $T_{qry}$, insert/update latency $T_{ins}$ (ms), and footprint $M$ (database size in MB). The **reference Pareto frontier** is computed from the same candidate pool using these measured metrics and is used only for evaluation.

*Inference-time cost reporting.* Offline profiling is used only for supervision and evaluation of ground truth and is *excluded* from all end-to-end runtime results. All reported runtimes in §5.3 measure only the inference-time pipeline (synthesis + concretization + prediction + Pareto/weighted selection).

*Baselines.* We compare against two established systems: *Trade-Maker* [11] and LEANT [9]. We additionally include two lightweight learning baselines trained on the same profiling labels: **Proxy-Only**, a tree-based regressor over static cost proxies $x_F(S)$, and **Feature-MLP**, a 2–3 layer MLP over flattened schema/workload statistics (e.g., counts of tables/columns/FKs/indexes, join-depth proxies, and workload template summaries aggregated by frequency). All learning-based methods predict $(T_{qry}, T_{ins}, M)$ and are evaluated under the same split and metrics as TRIORM.

*Training protocol (TRIORM).* We use leave-one-model-out evaluation to avoid leakage across closely related candidate schemas: for each of the nine models, we train on the other eight and evaluate on the held-out model; tables report per-model results and means across folds. We train a multitask regressor for $(T_{qry}, T_{ins}, M)$ using Huber loss on log-transformed targets. Unless stated otherwise, we freeze the schema/workload encoders and train only the fusion module and regression heads.

*Concretization validity and workload-frequency robustness.* We perform two checks to reduce the risk that conclusions depend on artifacts of concretization or frequency specification: (i) we validate concretization on a representative subset by comparing cost-relevant structure between ORM-logged SQL and our canonical templates (join count/graph, touched tables, write sets), and (ii) we perturb operation frequencies (±10%, ±25%, ±50%) and measure predicted Pareto-set stability (e.g., top-$k$ overlap / Jaccard). Recommendations remain stable under moderate frequency noise.

*Metrics.* For **RQ1**, we report *Generational Distance (GD)* (lower is better) and *Hypervolume (HV)* (higher is better) with respect to the measured reference frontier [19, 31]. For **RQ2**, we report Precision/Recall/F1 for Pareto membership prediction. For **RQ3**, we report end-to-end runtime and peak memory for the inference-time pipeline (excluding offline profiling).

*Hardware/DBMS.* We ran all experiments on a dedicated x86-64 machine with an AMD EPYC 7543P CPU, 32 physical cores (64 threads), 256 GB RAM, and NVMe SSD storage, running Ubuntu 22.04.3 LTS. We used PostgreSQL 14.10 with a tuned configuration: shared_buffers=64 GB;work_mem=64 MB;maintenance_work_mem=2 GB;effective_cache_size=192 GB;max_parallel_workers_per_gather=4. All methods use the same DBMS configuration.

## 5.3 Results

*RQ1: Pareto quality (GD/HV).* Table 2 reports GD/HV per model. Averaged across subjects, TRIORM improves GD (0.07→0.03) and

**Table 2: Per-system Pareto quality (GD/HV). Lower GD and higher HV are better.**

| Model | LEANT GD/HV | TRIORM GD/HV |
|---|---|---|
| Customer–Order | 0.07/0.61 | **0.03/0.81** |
| OnlineStore | 0.03/0.61 | **0.01/0.84** |
| BankMgmt. | 0.04/0.60 | **0.02/0.80** |
| Camping | 0.10/0.59 | **0.08/0.75** |
| Flagship | 0.06/0.62 | **0.03/0.78** |
| Decider | 0.08/0.63 | **0.00/0.85** |
| LibraryMgmt. | 0.05/0.60 | **0.01/0.78** |
| CSOS | 0.09/0.62 | **0.05/0.74** |
| E-commerce | 0.11/0.61 | **0.08/0.69** |
| **Mean** | **0.07/0.61** | **0.03/0.78** |

**Table 3: Pareto-set identification accuracy. Higher is better.**

| Method | Precision | Recall | F1 |
|---|---|---|---|
| LEANT [9] | 0.75 | 0.77 | 0.76 |
| **TRIORM** | **0.80** | **0.83** | **0.82** |

**Table 4: End-to-end efficiency (benchmark suite). Lower is better.**

| Method | Runtime (s) ↓ | Peak Mem. (GB) ↓ |
|---|---|---|
| TradeMaker [11] | $2.23 \times 10^6$ | 61.0 |
| LEANT [9] | $2.6 \times 10^4$ | 17.0 |
| **TRIORM** | $3.6 \times 10^3$ | **11.0** |

increases HV (0.61→0.78), indicating a predicted non-dominated set that is closer to and covers more of the measured frontier [19, 31]. Improvements are largest on models where mapping choices strongly affect join structure and write amplification (e.g., OnlineStore, LibraryMgmt, Decider).

*RQ2: Pareto-set identification (Precision/Recall/F1).* Table 3 reports Pareto membership classification quality. TRIORM improves over LEANT (0.75/0.77/0.76 → 0.80/0.83/0.82), indicating more accurate filtering of dominated candidates while retaining true Pareto-optimal designs.

*RQ3: End-to-end efficiency (runtime/memory).* Table 4 summarizes inference-time cost (excluding offline profiling). TRIORM reduces runtime from $2.6 \times 10^4$s to $3.6 \times 10^3$s and peak memory from 17.0GB to 11.0GB versus LEANT. Relative to TradeMaker, the reduction is orders of magnitude, reflecting the benefit of avoiding benchmark-heavy inner loops at recommendation time.

*RQ4: Input contribution and learning baselines.* Table 6 also reports prediction accuracy for each objective using mean absolute error (MAE) in log space for query latency, insert/update latency, and storage footprint. We study (i) TRIORM ablations to isolate the contribution of each input modality, and (ii) lightweight learning baselines trained on the same offline labels. For ablations, we compare the full tri-input model to single- and two-input variants, holding the training protocol and split fixed; inputs are removed by masking them with a learned null embedding (equivalently, zeroing after encoding). Table 6 shows that removing the concretized workload (I2) causes the largest degradation in GD/HV and Pareto-F1, confirming that mapping-induced SQL shape is critical for correct ranking; I1 remains important, and I3 provides a consistent robustness gain. Table 5 compares TRIORM with two lightweight predictors trained on the same profiling labels.

**Table 5: Learning baseline comparison (model-averaged). Lower GD and higher HV/F1 are better. Runtime excludes offline profiling.**

| Method | GD ↓ | HV ↑ | Pareto F1 ↑ | Runtime (s) ↓ |
|---|---|---|---|---|
| Proxy-Only ($x_F$) | 0.12 | 0.52 | 0.64 | $1.8 \times 10^3$ |
| Feature-MLP (flat feats) | 0.09 | 0.57 | 0.70 | $2.4 \times 10^3$ |
| Leant | 0.07 | 0.61 | 0.76 | $2.6 \times 10^4$ |
| **TriORM** | **0.03** | **0.78** | **0.82** | $3.6 \times 10^3$ |

**Table 6: Ablation study (model-averaged). Lower MAE/GD and higher HV/F1 are better. MAE is computed in log-space for latencies and footprint.**

| Variant | MAE $\log T_{\text{qry}}$ | MAE $\log T_{\text{ins}}$ | MAE $\log M$ | GD | HV | Pareto F1 |
|---|---|---|---|---|---|---|
| Full (I1+I2+I3) | 0.18 | 0.21 | 0.12 | 0.03 | 0.78 | 0.82 |
| I1+I2 (no I3) | 0.20 | 0.23 | 0.15 | 0.04 | 0.74 | 0.78 |
| I1+I3 (no I2) | 0.27 | 0.29 | 0.14 | 0.06 | 0.67 | 0.70 |
| I2+I3 (no I1) | 0.24 | 0.28 | 0.19 | 0.05 | 0.70 | 0.73 |
| I1 only | 0.30 | 0.32 | 0.16 | 0.07 | 0.63 | 0.66 |
| I2 only | 0.34 | 0.36 | 0.22 | 0.08 | 0.60 | 0.60 |
| I3 only | 0.38 | 0.40 | 0.25 | 0.10 | 0.55 | 0.55 |

## 6 DISCUSSION

*What drives the gains.* The consistent improvements in Pareto quality and Pareto-set identification (§5) are primarily driven by making *mapping-induced workload dependence* explicit. Even when two schemas are semantically valid, different inheritance and association encodings can yield different realized SQL (join topology, predicate placement, and write sets) [4, 12, 16, 21]. By jointly encoding (i) physical schema structure and (ii) the schema-concretized workload, TriORM reduces the systematic misranking that arises when models are conditioned on schemas alone.

*Role of cost proxies and interpretability.* While the schema and realized-workload encoders capture rich structural signals, lightweight cost proxies provide complementary benefits. First, they improve robustness across heterogeneous candidate sets by injecting compact indicators of key cost drivers (e.g., expected join depth, index coverage of predicates, and write-amplification proxies). Second, they support human-readable explanations: when two candidates trade query latency for insert/update cost, the system can attribute the tradeoff to concrete structural mechanisms (additional joins versus additional indexes/redundancy), consistent with established ORM performance guidance [4, 16].

*Intended usage and scope of "benchmark-free."* TriORM is intended for *design-time* decision support. Given $(O, \Phi)$ and an abstract workload (operation templates with approximate frequencies), it returns either (i) a small predicted Pareto set of mappings that are valid within the chosen synthesis bounds, or (ii) a single mapping via user-specified objective weights. In this work, "benchmark-free" refers strictly to the *online recommendation loop*: at recommendation time, TriORM does not perform per-candidate schema deployment, data loading, or workload execution. Offline profiling remains necessary to obtain supervision for training and to compute evaluation ground truth, consistent with benchmark-driven pipelines [9].

## 7 THREATS TO VALIDITY

*Construct validity.* We assess recommendation quality using GD and HV relative to a *measured* reference Pareto frontier, and report Pareto set Precision/Recall/F1 [19, 31]. These metrics can be sensitive to objective normalization and the HV reference point. To reduce metric-induced artifacts, we apply the same normalization and HV reference point across methods and compute the reference frontier from the same candidate pool per subject. We measure footprint as the on-disk database size after data loading and index creation; other notions of footprint (e.g., buffer-pool residency) could change absolute values, but are typically correlated with redundancy and index overhead.

*Internal validity.* Latency measurements may be influenced by system noise and implementation details (e.g., caching behavior, connection state, and aggregation choices). We mitigate these effects using a fixed profiling protocol across methods (warm-up, repeated runs, and consistent aggregation) and by running all experiments on a dedicated machine with a fixed DBMS configuration. For learning-based methods, outcomes can vary with hyperparameters and random seeds; we use a split-by-model protocol to avoid train/test leakage across closely related schemas and report model-averaged results over multiple seeds.

*External validity.* We evaluate on the standard TradeMaker/Leant benchmark suite [9, 11] under PostgreSQL 14, which enables direct comparability with prior work. Generalization may change under substantial workload shifts (e.g., new templates or different frequency mixes) or when moving to different DBMS engines and configurations.

*Bounded synthesis and design-space coverage.* Alloy-based synthesis operates under finite scopes [20]. Accordingly, TriORM optimizes over the bounded candidate space induced by the chosen scopes and constraints $\Phi$, rather than the unbounded space of all possible ORM mappings. We therefore interpret results as tradeoff quality *within* this bounded design space and report synthesis statistics and candidate counts for transparency.

## 8 RELATED WORK

*ORM mapping patterns and performance tradeoffs.* Foundational work on ORM design has shown that inheritance strategies (e.g., single-table, joined-subclass, and table-per-class) and association encodings (e.g., foreign keys versus join tables) can substantially affect join depth, redundancy, integrity enforcement, and update behavior, thereby shaping latency–footprint tradeoffs [4, 12, 16, 21]. These studies motivate reasoning over multiple semantically equivalent mappings rather than relying on fixed ORM defaults.

*Correctness-preserving schema synthesis.* Specification-based approaches encode mapping rules and integrity constraints in relational logic and synthesize only schemas that satisfy those constraints by construction [11, 20]. This line of work provides strong semantic guarantees, but does not by itself resolve how to efficiently rank large numbers of valid candidates under workload-specific performance objectives. TriORM builds on this foundation by preserving validity through bounded Alloy synthesis while shifting the main emphasis to workload-aware ranking within the synthesized design space.

*Benchmark-driven exploration for ORM optimization.* Prior ORM optimizers attach measured performance to candidate schemas by repeatedly deploying schemas, loading data, and executing workloads across the candidate space [9]. Although accurate, this inner-loop benchmarking cost quickly becomes the dominant bottleneck as the number of valid candidates grows, limiting interactive use in larger synthesis spaces [10, 12, 15, 18, 21, 25, 29]. TRiORM retains offline measurement for supervision, but removes benchmarking from the inference-time decision loop by predicting continuous objectives from both schema structure and the mapping-induced workload.

*Learning-based ORM optimization and companion work.* Recent learning-based approaches seek to accelerate ORM design exploration by predicting promising candidates from schema-centric representations, often emphasizing classification or coarse ranking [17]. For example, DTS applies Transformer-based models to identify near-Pareto schema candidates primarily from schema features [17]. A preliminary ICSE 2026 extended abstract introduced early ideas behind Y-Map as a performance-aware learning framework for ORM design [7], and a subsequent AIware 2026 short paper developed this direction further through neural–symbolic optimization for performance-aware ORM database design [8]. TRiORM extends this line of work by explicitly modeling workload-conditioned performance: rather than reasoning from schema structure alone, it concretizes the abstract workload for each valid mapping and incorporates the realized SQL workload into a tri-input continuous predictor. In parallel, learned models have also been used for workload-aware decisions in database systems, including learned query optimization and ML-driven tuning [3, 23]. TRiORM complements these efforts by targeting ORM mapping spaces specifically and by coupling workload-aware prediction with correctness-preserving synthesis.

*Neural representations for schemas and SQL..* Graph neural networks for multi-relation graphs, such as R-GCN, provide principled encodings of typed dependencies, including PK/FK links and inheritance relations [27], while Transformer encoders are effective backbones for modeling SQL templates as structured sequences [30]. TRiORM combines these strands in a tri-input design that integrates schema graphs, concretized workloads, and static cost proxies, enabling workload-sensitive prediction beyond schema-only embeddings.

TRiORM extends this line of work by explicitly modeling *workload-conditioned* performance: rather than reasoning from schema structure alone, it concretizes the abstract workload for each valid mapping and incorporates the realized SQL workload into a tri-input continuous predictor.

## 9 CONCLUSION

We presented TRiORM, a workload-aware neural–symbolic framework for multi-objective ORM mapping design. TRiORM enforces correctness-by-construction by synthesizing only admissible mappings from a bounded relational specification, then ranks valid candidates using a tri-input predictor that models (i) typed schema structure, (ii) the *schema-concretized* SQL workload induced by each mapping, and (iii) compact static proxies capturing join and write

pressure. This design targets the central challenge in ORM optimization: semantically equivalent mappings can induce different SQL shapes and physical cost drivers, so accurate ranking must be conditioned on both the schema and its realized workload. Across nine benchmark object models from the TradeMaker/LEANT suite, TRiORM improves Pareto-front approximation and Pareto-set identification while reducing inference-time optimization cost relative to representative synthesis- and benchmarking-driven baselines. Overall, our results support a practical direction for performance-aware ORM tooling: use symbolic methods to guarantee validity and systematically explore the design space, and use learned, workload-conditioned prediction to enable fast tradeoff exploration without per-candidate benchmarking in the online loop.

## ACKNOWLEDGMENT

The authors gratefully acknowledge the support of Bellevue University for this research.

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
