# OpenReview forum: "TriORM: Workload-Aware Neural--Symbolic Multi-Objective Optimization for ORM Mapping Design"
_ACM.org/AIWare/2026/Conference — AIware 2026_

### Official Review · Reviewer_F6JF · 2026-02-25

**Rating:** 3
**Confidence:** 2

**Review:**

### **Strengths**

- **Real Problem:** The workload concretization step directly targets a real pain point: semantically equivalent mappings can induce different SQL shapes/cost drivers, so ranking should be conditioned on the realized workload, not schema alone.
- **Well-structured Design:** The design logics is clear: symbolic synthesis for validity, learned prediction for performance ranking, then Pareto/weighted selection.


### Comments

1. **Missing comparison to the strongest recent  baseline (DTS/FSE’25) leaves positioning unclear.**

   The paper primarily compares against Leant, but does not include DTS, the most recent baselines (but it is cited as [13]). I would recommend authors to add a DTS comparison under the same candidate pool and metric definitions, or clearly explain why it cannot be included.

**Summary:**

TriORM proposes a workload-aware neural–symbolic pipeline for multi-objective ORM mapping design. It (1) enumerates only admissible mappings via bounded relational synthesis (Alloy), (2) concretizes an abstract workload sketch (operation templates + frequencies) into schema-specific SQL/transaction templates per candidate, making mapping-induced SQL differences explicit and comparable, and (3) predicts continuous objectives using a tri-input model that fuses a typed schema-graph encoder, a concretized-workload encoder, and compact static cost proxies, that enabling Pareto filtering or user-weighted selection without per-candidate execution in the recommendation loop.

---

> ### Author Response · Authors · 2026-03-14
> **Response to Reviewer F6JF**
>
> Thank you for the thoughtful review and for the positive assessment of the problem setting and the overall design. We especially appreciate your recognition that workload concretization is important for ORM mapping design, since semantically equivalent mappings can induce materially different SQL structures and cost behavior.
>
> We agree that comparison to a strong recent baseline, such as **DTS**, would help further clarify positioning. In the current paper, we centered the evaluation on the **TradeMaker/Leant benchmark suite** and compared against the most directly comparable baselines available in that same controlled setting, namely **TradeMaker** and **Leant**, as well as two lightweight learning-based baselines trained on the same profiling labels. Our goal was to keep the **candidate pool, workload templates, DBMS setting, and evaluation metrics fixed** so that differences could be attributed to the recommendation method rather than to changes in evaluation setup.
>
> DTS is cited in the paper as a relevant recent learning-based direction. Our intent in citing it was to position TriORM relative to recent schema-centric learning approaches that improve scalability by predicting promising candidates. The key methodological distinction is that **TriORM predicts continuous multi-objective performance while explicitly conditioning on the schema-concretized workload induced by each mapping**, rather than relying only on schema-level representations. This distinction is also supported by our ablation results, which show that removing the concretized workload input yields the greatest degradation in predictive and recommendation quality.
>
> We did not include a DTS comparison in the current version, and we agree that this should have been explained more clearly. The present evaluation was scoped to methods that could be compared under the same benchmark suite and measurement protocol used throughout the paper. We agree that adding a direct DTS comparison under the same candidate pool and metric definitions would further strengthen empirical positioning, and we will make the scope and rationale of the current baseline selection clearer in the paper.

---

### Official Review · Reviewer_MHvR · 2026-03-03

**Rating:** 4
**Confidence:** 4

**Review:**

The idea of using a model to predict the runtime metrics is very interesting and practical.
The results also show improvements on multiple fronts.

One part that is not clear is how the workload is realized. The paper only mentions that this is a deterministic process, but how the frequency of operations is determined is not clear.

**Summary:**

This paper presents TriORM, which aims to automate ORM without deploying and profiling any mapping.
The main idea is to train a model to predict metrics from the typed schema graph, and the realized workload. This allows evaluating a mapping without running a profiler.

Evaluation on a set of 9 systems shows improvements over the baselines in terms of accurately identifying the pareto set, GD/HV, time, and peak memory.

---

> ### Author Response · Authors · 2026-03-14
> **Response to Reviewer MHvR**
>
> Thank you for the positive and encouraging feedback. We appreciate your recognition that the predictive model is both practical and central to the paper's contribution. We also agree that clarifying the model’s standalone accuracy and the workload realization process would strengthen the presentation.
>
> **1. Fine-grained predictive accuracy of the model.**
> We agree that reporting the predictor’s standalone accuracy is important. For clarity, **Table 6** reports the model’s fine-grained predictive accuracy using model-averaged MAE values in log-space for query latency, insert/update latency, and storage footprint. For the full TriORM model, these errors are **0.18**, **0.21**, and **0.12**, respectively. We agree that these results are useful both for interpreting the current method and for identifying where future improvements are most needed. In addition, the ablation results show that removing the concretized workload input causes the greatest degradation in predictive and recommendation quality, helping localize the main source of gain in the full model.
>
> **2. Clarification of workload realization and operation frequencies.**
> The workload realization process starts with an abstract workload specification consisting of operation templates and their associated frequencies. For each candidate schema, this abstract workload is deterministically concretized into schema-specific SQL and transaction templates. The purpose of this step is to preserve the same logical workload intent while making mapping-induced differences in realized SQL explicit.
>
> The operation frequencies are therefore not inferred during concretization; they are taken directly from the benchmark workload templates and carried into the schema-specific realized workload. Because frequency specification can strongly influence recommendation quality, we also included a robustness check in which operation frequencies were perturbed by **±10%**, **±25%**, and **±50%**, and recommendation stability was then measured. Recommendations remained stable under moderate frequency noise. We agree that this part should have been stated more explicitly in the paper.
>
> We thank the reviewer again for the helpful comments. We believe these clarifications make both the predictive component and the workload realization process more transparent, and we will make these points clearer in the paper.

---

> > ### Comment · Reviewer_MHvR · 2026-03-14
> >
> > Thank you for the clarifications.

---

### Official Review · Reviewer_c4ir · 2026-03-12

**Rating:** 2
**Confidence:** 2

**Review:**

I am not an expert on database systems and the related literature on ORM optimization work. Conceptually the work seems novel, the use of Alloy to generate correct by construction schemas but also condition them to workload type and generated predictions. Using a fused representation of schema, workload and cost to generate a continuous prediction.

The paper is generally written well, the core idea and technical details are presented well. I would suggest the authors elaborate on Algorithm-1 more, it is not clearly described. For example, it is not clear what the functions Graphify, Concretize, Trans, Synth actually entails. I can infer some semantics based on the algorithm, but it is recommended to clearly define all methods and assumptions of the algorithm such that another researcher can implement the same algorithm. What does alpha, beta and gamma weights capture? How are they determined? Did the authors define them in their implementation? Is there any evaluation of how the performance varies with different weights?

It is not clear how the authors train the regression model, the description is very vague. How did the authors train the model and what data was used are unclear. Please elaborate more on the details of the training phase as this is one of the core contributions of this work, the fusion and regression.

The evaluation setup does not describe the selection criteria for the dataset and baselines. Why did the authors select old benchmarks from 2017, are there not recent benchmarks curated? What is the selection criteria for excluding other benchmarks? What is the selection criteria for baselines? Are there other baselines that are excluded from comparison? If so, why?

**Summary:**

This paper presents a workload aware framework named TriORM, that combines symbolic constraints guided synthesis for PRM schema generation and neural network to predict the workload associated with a candidate schema. Evaluation using 9 object models from prior work, shows the proposed technique out performs considered baselines.

---

> ### Author Response · Authors · 2026-03-14
> **Response to Reviewer c4ir**
>
> We appreciate your positive assessment of the framework's conceptual novelty and your helpful suggestions for improving clarity. We agree that some implementation details—especially those related to Algorithm 1, the training procedure, and the evaluation setup—could have been stated more explicitly in the short paper format. We clarify each point below.
>
> **1. Clarification of Algorithm 1 and the functions used in it.**
> Algorithm 1 summarizes the end-to-end pipeline that combines symbolic synthesis with learned performance prediction.
>
> * **Trans(O, Φ)** transforms the input object model (O) together with the admissibility constraints (Φ) into a bounded Alloy specification. This specification defines the valid ORM mapping design space while enforcing structural and integrity constraints.
> * **Synth(A)** uses the Alloy analyzer to enumerate candidate ORM schemas that satisfy the specification (A). Since these schemas are generated from the constraint system, all candidates are valid by construction.
> * **Concretize(Wabs, Si)** instantiates the abstract workload specification for a specific candidate schema (S_i). The abstract workload contains operation templates together with their frequencies, and concretization produces schema-specific SQL and transaction templates. This is important because semantically equivalent mappings can induce different SQL structures and cost behavior.
> * **Graphify(Si)** converts the candidate relational schema into a typed schema graph used by the schema encoder. In this graph, nodes represent schema elements, such as relations and attributes, and edges capture structural dependencies, such as foreign-key relationships.
>
> These steps produce the three inputs used by the predictive model:
> (i) the typed schema graph,
> (ii) the concretized workload representation, and
> (iii) compact static cost proxies.
>
> **2. Meaning of the alpha, beta, and gamma weights.**
> The weights are **not learned parameters of the regression model**. They are user-specified preference weights used only during the **selection stage** after prediction. Their purpose is to express deployment-time trade-offs among the predicted objectives, such as query latency, insert/update latency, and storage footprint. When these weights are provided, the system selects the schema that minimizes the weighted objective. When no preference weights are provided, the system instead returns the Pareto set of non-dominated schemas. The main evaluation in the paper focuses on Pareto recommendation quality rather than sensitivity to specific weight settings. We agree that this distinction should have been made more explicit.
>
> **3. Clarification of how the regression model is trained and what data is used.**
>
> The training dataset is constructed as follows. First, valid ORM schemas are generated through the synthesis stage. Each candidate schema is then materialized in PostgreSQL. Next, the concretized workload for that schema is executed under a controlled profiling protocol to obtain ground-truth measurements for the target objectives: query latency, insert/update latency, and storage footprint. Across the benchmark suite, this yields approximately 9,500 valid schema instances with measured objective values.
>
> The predictive model is trained to estimate these objectives from the fused inputs consisting of the typed schema graph, the concretized workload representation, and compact cost proxies. We use **leave-one-model-out cross-validation** so that schemas derived from the same object model do not appear in both the training and test folds, thereby reducing leakage across closely related candidates. The model is trained as a **multitask regressor** on **log-transformed targets** using **Huber loss**. Unless otherwise stated, the schema and workload encoders are frozen, and only the fusion module and regression heads are trained. We agree that these details should have been presented more clearly in the paper.
>
> **4. Dataset and benchmark selection.**
> We use the benchmark suite commonly adopted in prior ORM mapping optimization research because it provides a controlled, reproducible, and directly comparable evaluation setting. It includes nine object models with associated workloads and remains the standard reference suite in this area. Our goal was comparability with prior work rather than claiming the benchmarks are the newest available.
>
> **5. Baseline selection and possible exclusions.**
> The baselines were chosen to represent the most directly relevant approaches for this problem setting. We include established ORM optimization methods in the same benchmark setting, together with two lightweight learning-based baselines trained on the same profiling labels. This provides a controlled comparison under the same candidate pool, DBMS setting, and evaluation metrics.
>
> Our goal was to compare against the most directly comparable baselines rather than exhaustively include every prior method. We agree that this point should have been clearer.

---

### Author Response · Authors · 2026-03-14
**Global Author Response**

We thank all reviewers for their thoughtful and constructive feedback. We appreciate the positive assessment of the paper’s core idea and the helpful suggestions for improving clarity, reproducibility, and empirical positioning.

Across the reviews, four main points merit clarification.

**First, regarding the overall pipeline and Algorithm 1**, TriORM combines:
(1) constraint-guided synthesis to enumerate only admissible ORM mappings,
(2) deterministic concretization of an abstract workload into schema-specific SQL/transaction templates for each candidate mapping, and
(3) a tri-input predictive model that fuses the typed schema graph, the concretized workload, and compact static cost proxies to estimate continuous objectives for recommendation.

**Second, regarding the predictive model**, the paper reports the model’s fine-grained predictive accuracy in **Table 6** using model-averaged MAE values in log-space for the three objectives. For the full TriORM model, the reported errors are **0.18** for query latency, **0.21** for insert/update latency, and **0.12** for storage footprint. The ablation results further show that removing the concretized workload input leads to the largest degradation, underscoring the importance of workload-aware prediction.

**Third, regarding workload realization**, TriORM begins from abstract operation templates together with their associated frequencies. For each candidate schema, this workload is deterministically concretized into schema-specific SQL and transaction templates while preserving the same logical workload intent. The operation frequencies are taken directly from the benchmark workload templates and carried into the realized workload. We also evaluated robustness under perturbed operation frequencies to assess recommendation stability.

**Fourth, regarding evaluation scope**, the experiments use the benchmark suite commonly adopted in prior ORM mapping optimization research because it provides a controlled, reproducible, and directly comparable setting. The baseline selection was guided by the same principle of comparability under a common candidate pool, workload setting, DBMS configuration, and evaluation protocol.

We agree that several of these points could have been stated more explicitly in the short-paper format. We are grateful for the reviewers’ comments and have addressed each point in more detail in our individual responses.